# Azobenzene Functionalized “T-Type” Poly(Amide Imide)s vs. Guest-Host Systems—A Comparative Study of Structure-Property Relations

**DOI:** 10.3390/ma13081912

**Published:** 2020-04-18

**Authors:** Karolina Bujak, Anna Kozanecka-Szmigiel, Ewa Schab-Balcerzak, Jolanta Konieczkowska

**Affiliations:** 1Institute of Chemistry, University of Silesia, Szkolna 9, 40-006 Katowice, Poland; bujak.karolina@outlook.com; 2Faculty of Physics, Warsaw University of Technology, 75 Koszykowa Str., 00-662 Warszawa, Poland; anna.szmigiel@pw.edu.pl; 3Centre of Polymer and Carbon Materials, Polish Academy of Sciences, 34 M. Curie-Sklodowska Str., 41-819 Zabrze, Poland; ebalcerzak@cmpw-pan.edu.pl

**Keywords:** azobenzene, azo polymers, poly(amide imide)s, photoinduced birefringence

## Abstract

This paper describes the synthesis and characterization of new “T-type” azo poly(amide imide)s as well as guest-host systems based on the “T-type” matrices. The matrices possessed pyridine rings in a main-chain and azobenzene moieties located either between the amide or imide groups. The non-covalent polymers contained the molecularly dispersed 4-phenylazophenol or 4-[(4-methyl phenyl)diazinyl]phenol chromophores that are capable of forming intermolecular hydrogen bonds with the pyridine rings. The FTIR spectroscopy and the measurements of the thermal, optical and photoinduced optical birefringence were employed for the determination of the influence of H-bonds and the specific elements of polymer architecture on physicochemical properties. Moreover, the obtained results were compared to those described in our previous works to formulate structure-property relations that may be considered general for the class of “T-type” azo poly(amide imide)s.

## 1. Introduction

One of the fastest-growing fields of technologies is photonics and optoelectronics. The growing interest in photonic processes has become the propelling force of research into new materials that may be applied for information storage, as optical elements, controlled optical and photo-optical media, as light couplers in planar waveguides, liquid-crystal layering in liquid crystal displays, etc. [1,2,3,4,5,6,7,8,9,10,11,12,13,14]. One of the most attractive materials for photonics applications are azo polymers i.e., macromolecules containing chromophores being derivatives of azobenzene or azo pyridine [15,16]. Polymeric materials can be easily adapted to meet specific applications, due to their advantages such as flexibility, ease of processing, low dielectric constant and their stability at a set temperature [17]. Depending on the kind of incorporation of azo chromophores to the polymer backbone, azo polymers can be classified as functionalized with a covalently bonded azo chromophore (side-chain, main-chain and “T-type” polymers), or guest-host azo systems, within azo dye, which is dispersed in the polymer (doped polymers) or attached to the polymer matrix by non-covalent interactions as H-bonds, ionic bonds or π–π sticks (supramolecular assembles) [3,18,19].

All potential applications of azo polymers result from light-stimuli properties. The irradiation of azo polymers with linearly polarized light forces multiple cycles of trans-cis-trans isomerization of azobenzene molecules, which is preceded by selective absorption events. The reorientation process of azo chromophores towards positions perpendicular to the light polarization (in which the azo molecules cannot be photoexcited) leads to the generation of the so-called photoinduced optical anisotropy (POA). The photoinduced optical anisotropy is observed as a difference in the refractive light index (birefringence) and/or the absorption coefficient (dichroism) experienced by light polarized in a direction perpendicular and parallel to the polarization of the excitation beam, respectively [17,20]. Due to the fact that POA is influenced by many factors referring to the chemical structure of azo polymers, a detailed characterization of the relationship between azo polymer architecture and the photoinduced effects is required for the optimization of properties for specific uses.

Polyimides (PIs) are an important group of polymers among the materials dedicated to photonics and optoelectronics applications, due to their unique properties i.e., low susceptibility to laser light damage, high glass-transition temperature and thermal stability [21]. Functionalized [8,22,23,24] and non-covalent azo polyimides [25,26,27,28] were widely investigated in the last decade for the generation of photoinduced birefringence or surface relief gratings. However, only one work showed mixed azo systems with two azobenzenes, where one was covalently bonded to the polymer backbone and the second was dispersed in the PI matrix [29]. The azo polyimide matrix containing chromophores with the nitro group was doped by two azo dyes, disperse red 1 or azo-carbazole diamine. Photoinduced birefringence was ca. 0.035 for polymer matrix and 0.062 and 0.07 (argon laser, λ = 488 nm, t = 1500 s) for mixed azo systems with disperse red 1 and azo-carbazole diamine, respectively. The enhancement of birefringence was caused by the increased content of azo chromophores in the polymer. However, the relaxation process after turning off the laser beam was significantly lower for the azo system doped by azo-carbazole diamine than the polyimide matrix. Polyimides are attractive hosts for the preparation of azo materials exhibiting a high and stable optical anisotropy [17]. The guest-host azo polyimides generate large modulations of surface relief gratings up to 95 nm and high birefringence up to 0.035 with 20% relaxation after 10 min [30,31]. Additionally, it was proved that materials exhibiting birefringence at ca. 0.02 may be applied in photonic devices [32].

Our previous works showed that poly(amide imide)s with two azobenzene moieties per structural unit exhibited the highest and the most stable photoinduced birefringence, as was observed for polyimides [33]. On the other hand, poly(amide imide)s with phenyl rings in the main-chain and azobenzene between the imide structure did not exhibit birefringence, because of the forming intermolecular hydrogen bonds; in addition, poly(amide imide)s with chromophores between amide linkages were insoluble in organic solvents [34,35]. The unexpected results for PIs with one azo group in the polymer-repeating unit made it impossible to research the influence of a location of an azo dye (between imide or amide linkages) on the physicochemical properties, including the photoinduced birefringence. Our previous investigations became an inspiration to the preparation of new poly(amide imide)s with pyridine rings in the main-chain, which improved the solubility. Functionalized PIs were applied as matrices for the preparation of the non-covalent azo systems with two azobenzene moieties per repeating unit, where one azo chromophore was covalently bonded to the polymer backbone and the second was dispersed molecularly in the polymer matrix. The prepared azo polyimides allowed for the study of the influence of the azo dye content and their location in the backbone on the physicochemical properties in comparison to the PIs with covalently bonded azo chromophores or fully non-covalent azo systems.

## 2. Results and Discussion

In this paper, two series of “T-type” azo polyimides i.e., functionalized and guest-host azo systems (Figure 1) are described. Functionalized poly(amide imide)s **PAI-5**–**PAI-8** and doped azo system **PAI-9[Az(CH_3_)]** were synthesized and characterized in our previous works [33,34,35,36,37]. Polyimides with pyridine rings in the polymer backbone are new materials. Polymers **PAI-1**, **PAI-2** and **PAI-4** contain azobenzene moiety between amide groups, while **PAI-3**, **PAI-5,** and **PAI-6** have azobenzene unit between imide rings. Polymers **PAI-7** and **PAI-8** have two covalently bonded azobenzenes between amide and imide groups. The functionalized poly(amide imide)s with pyridine rings in the main-chain were applied as matrices to obtain the guest-host azo systems. Azo chromophore 4-phenylazophenol (**Az(H)**) was dispersed in the **PAI-1** and **PAI-3** matrices, while 4-[(4-methyl phenyl)diazinyl]phenol (**Az(CH_3_)**) was added to the **PAI-2** and **PAI-9** polymers. The guest-host azo systems contained two chromophores (with the same substituent –CH_3_ or H), the first one covalently bonded to the backbone, and the second dispersed molecularly in a polymer matrix. Non-covalent azo polyimides were prepared by dissolution equimolar amounts of the polymer matrix and the azo chromophore. Only the polyimide **PAI-9[Az(CH_3_)]** contained two non-covalently bonded azo chromophores. The chemical structures of the investigated azo polymers are shown in Figure 1.

### 2.1. Intermolecular Hydrogen Bonds Formation

^1^H NMR and FTIR spectroscopies are powerful techniques to prove the hydrogen bond formation. In the ^1^H NMR spectra, the chemical shift of the amide proton can provide information on the H-bonds formation. It was found that if the hydrogen bonds became stronger the shift of signals was moved downfield [38]. We proved that poly(amide imide)s containing covalently bonded azobenzene between imide rings (**PAI-5** and **PAI-6**) and without side azo groups readily formed the intermolecular hydrogen bonds between amide linkages [34,36]. The formation of the intermolecular H-bonds by polymers significantly influenced their physical properties i.e., led to poor solubility in organic solvents, generated a large blue shift of the absorption maximum, or hampered the photoinduced birefringence [34,36].

The presence of the intermolecular H-bonds for functionalized azo polyimide matrices was verified using FTIR and ^1^H NMR spectra. The comparison between the normalized spectra showed that the N–H absorption region at ca. 3300 cm^−1^ is higher in intensity and broaden for **PAI-3** (Figure 2a). Similarly, the vibration band of the C=O bond in the amide group of **PAI-3** was shifted to lower frequencies (at ca. 1650 cm^−1^) in comparison to **PAI-1** and **PAI-2** (Figure 2b). The shifts suggest the presence of interchain H-bonds in **PAI-3**. The presence of the intermolecular H-bonds for the functionalized azo polyimides with pyridine rings in the main-chain was verified by comparing the N–H signal position in the ^1^H NMR spectra taken at room temperature and at 80 °C. The chemical shift of the amide protons from ca. 11.35, 11.05 ppm to 11.30 and 10.90 ppm with the increasing temperature confirmed the presence of H-bonds, which became weaker at higher temperatures for all azo polyimide matrices. ^1^H NMR spectra of **PAI-2** registered at room temperature and at 80 °C are depicted in Figure 3. FTIR and ^1^H NMR spectra showed that intermolecular H-bonds are formed in azo polyimide matrices, but that the strongest hydrogen interactions were observed for **PAI-3**. A lack of azo moiety between amide groups favored the formation of H-bonds.

The incorporation of pyridine rings in the main-chain of the polymer is commonly used to –N∙∙∙H– hydrogen bonds formation between the polymer and azo chromophore in the supramolecular azo systems [9,11,39,40,41]. However, we proved that carbonyl groups in imide rings or amide linkages may hinder the formation of the H-bonds [25,36]. The ^1^H NMR spectra confirmed that the azo chromophores were dispersed in a polymer matrix without H-bonds formation. No reduction in signal intensity corresponding to the hydroxyl group in the chromophore structure was observed, and no signal shift corresponding to protons in the pyridine ring at ca. 8.9 ppm was detected either (Figure 4). The FTIR spectra exhibited the same character for the polymer matrix and the non-covalent azo system in the regions corresponding to the vibration of the N–H-bond in the amide group (Figure 5a), and characteristic for the absorption of pyridine ring (Figure 5b). Thus, non-covalent polyimides should be treated as doped systems, where azo chromophores are dispersed in the polymer matrix with no H-bonds formation. Simultaneously, broad and high-intensity N–H bands centered at ca. 3200 cm^−1^ (Figure 5a) could be the result of the presence of OH groups attached to the azo chromophore, dispersed in the polymer matrix, and of the formation of the interchain hydrogen bonds between the amide groups in both the **PAI-3** polymer matrix and its non-covalent analogue. We recently found an analogous effect for structurally similar azo poly(amide imide)s containing the chromophores between the amide groups and phenyl rings instead of the pyridine ones in the backbone [34]. The absorption band observed at 850 cm^−1^ in the FTIR spectrum of **PAI-3[Az(H)]** in Figure 5b was probably due to C–OH vibration in the azobenzene derivative **Az(H)**. This band was observed in the FTIR spectra of both chromophores (**Az(H)** and **Az(CH_3_)**) and in systems with dispersed **Az(H)** and **Az(CH_3_)** in a polymer matrix.

### 2.2. Polymer Characterization 

The polymer structures were determined by ^1^H NMR and FTIR spectroscopies and elemental analysis. The ^1^H NMR spectra showed signals in the range 6.89–8.87 ppm corresponding to the aromatic rings and at ca. 11.00 ppm, characteristic for the proton in the amide group. In the ^1^H NMR spectra of **PAI-2**, a signal at 2.38 ppm corresponding to the methyl group was observed. The FTIR spectra exhibited absorption bands of the imide rings at ~1780 cm^−1^, and ~1720 cm^−1^ attributed to the asymmetric and symmetric stretching vibration of the carbonyl group in the five-membered ring, respectively. The absorption peaks at 1356 cm^−1^ (**PAI-1** and **PAI-2**) and 1380 cm^−1^ (**PAI-3**) and ~720 cm^−1^ were attributed to C–N stretching and the deformation vibration of the imide ring, respectively. Poly(amide imide)s exhibited a broad absorption band in the range 3341–3151 cm^−1^, corresponding to the vibration of the N–H-bond, at 1679 cm^−1^ (**PAI-1** and **PAI-2**) and 1658 cm^−1^ (**PAI-3**), characteristic for C=O in the amide group. Polymer matrices exhibited absorption at ca. ~1100 cm^−1^ attributed to the pyridine ring. Elemental analysis showed a good correlation between the calculated and found values of the element content. The highest differences were 0.73% for C, 0.56% for N, and 0.49% for H.

The poly(amide imide)s structures were evaluated by the wide-angle X-ray diffraction measurements. All functionalized and guest-host azo polyimides showed the same diffraction patterns with one broad diffraction peak of the diffusion type in the range 17–38° (2θ). This was typical for perfectly amorphous materials. The selected X-ray patterns are presented in Appendix A. The solubility of the poly(amide imide)s was determined qualitatively by the dissolution of 10 mg of the polymer in 1 mL organic solvent. All poly(amide imide)s were soluble in N–methyl–2–pyrrolidone and partially soluble in N,N–Dimethylformamide (DMF), DMSO, CHCl_3_, THF and cyclohexanone, after heating to the boiling temperature. The exception was **PAI-1** which was soluble at room temperature in DMSO. Poly(amide imide)s exhibited a rather poor solubility in organic solvents, but the incorporation of the pyridine ring to the polymer backbone improved the solubility in comparison to the polyimides with phenyl rings (**PAI-4**, **PAI-5,** and **PAI-6**). The incorporation of the pyridine rings to the main-chain increased the solubility, which was also observed for different polyimides [42,43,44,45,46]. Poly(amide imide)s **PAI-5** and **PAI-6,** with azobenzene moieties between imide groups and phenyl rings located between amide linkages, were insoluble in THF and CHCl_3_. Polyimide **PAI-4** with phenyl rings between imide groups were insoluble in all the used solvents, because of physicochemical properties that were not investigated.

In Table 1 the average molar masses of poly(amide imide)s matrices are collected. The obtained values of molecular masses should be treated only indicatively. The molar masses calculated based on the calibration may differ from the absolute molar masses if the hydrodynamic values of the studied polymers differ from those of the polystyrene standards. The gel permeation chromatography analysis revealed the average weight of the molar masses (*M_w_*) in the range of 1.7 × 10^3^–14.9 × 10^3^ g/mol with a relatively low dispersity from 1.5 to 1.9. It is worth noting that the polymers with the azobenzenes between the imide rings (**PAI-3**, **PAI-5**, and **PAI-6**) exhibited the lowest *M_w_*. The *M_w_* values suggest the oligomeric nature of the polyimides.

### 2.3. Thermal Properties

The investigations of the glass transition temperatures of the functionalized poly(amide imide)s, the azo chromophores and the guest-host azo systems were evaluated by the differential scanning calorimetry (DSC) technique. The utilization of high-boiling temperature solvent, i.e., N-methyl-2-pyrrolidone (NMP), required a proper drying procedure (cf. caption 3.8) as the presence of the residual solvent in a material sample might lead to unreliable results. Table 2 summarizes the glass transition temperatures (*T_g_*) values of the functionalized poly(amide imide)s, chromophores and guest-host azo systems.

Functionalized poly(amide imide)s exhibited high *T_g_*s in the range of 244–309 °C. Considering the influence of the elements of the polymer structure on the *T_g_*, it was found that:Polymer with pyridine rings in the backbone exhibited higher *T_g_* than their analogue with phenyl rings (**PAI-3** vs. **PAI-5**).The incorporation of the azobenzene group between the amide linkages decreased the *T_g_* (**PAI-1**) in comparison to **PAI-3** containing chromophores between imide rings.The incorporation of two covalently bonded azo chromophores to the polymer repeating unit did not influence *T_g_*’s in comparison to the polymer with one azo group per structural unit, despite a significantly higher average for molar masses.The guest-host azo systems were characterized by lower *T_g_* in comparison to polymer matrices, because of the plastification effect. A much greater reduction of *T_g_* was observed for **PAI-9[Az(CH_3_)]** than for the other guest-host systems due to the higher content of non-covalently bonded azo dye (25 wt.% vs. 45 wt.%).

The thermal stability of the poly(amide imide)s matrices and guest-host azo systems were investigated by the thermogravimetric (TGA) measurements and the results are collected in Table 2. The beginning of thermal decomposition for functionalized poly(amide imide)s based on temperatures of the 5% (*T_5_*) and 10% (*T_10_*) weight loss appeared in the range of 258–332 °C and 395–412 °C, respectively. Poly(amide imide)s matrices showed two or three decomposition steps with a temperature of the maximum decomposition rate (*T_max_*), as evidenced by the differential thermogravimetric curves (DTG), in the range of 275–606 °C (Appendix A). The first and second degradation step with the weight loss in the range 275–473 °C was connected with the destruction of the azo groups, while the third step in the range 546–606 °C was due to the degradation of the polymer backbone [48,49]. The incorporation of the pyridine ring to the polymer main-chain did not influence thermal stability in comparison to the polyimides with phenyl rings. All functionalized polyimides were characterized by a high residual weight at 800 °C, ca. 50%, despite **PAI-8**.

In the next stage, the effect of the presence of a non-covalent azo dye in azo systems on thermal stability, as compared to the stability of the host polymer, was examined. It was found that the incorporation of the chromophore into the polymer matrix lowered the *T_5_* by 130–191 °C and *T_10_* by 170–223 °C (Table 2). The highest decrease of the beginning of the thermal decomposition temperature was observed for polyimide **PAI-9[Az(CH_3_)]** with 45 wt.% of the chromophore. The presence of azo dye caused a slight drop in the temperature of the maximum decomposition rate in azo systems compared to the polymeric hosts, and reduction of residual weight in the range of 6–29%. Guest-host polymers showed two or three decomposition steps in the range of 172–581 °C. The first degradation step with the weight loss between 172–226 °C was connected with the destruction of a non-covalently bonded azo chromophore, the second step from 373–420°C was connected with the degradation of N=N linkages in covalently attached azo moieties, while the third step (the second step for **PAI-3[Az(CH_3_)]**) in the range of 362–583 °C was due to the degradation of the polymer main-chain.

### 2.4. Linear Optical Properties

The UV-Vis spectra of the polymer matrices were acquired both in polymer films cast on a glass substrate and in an NMP solution. The range of UV-Vis measurements was limited by the transparency of the used substrate and the solvent. The maximum absorption wavelengths of studied polymers are collected in Table 3.

Functionalized polyimides are characterized by the maximum of the absorption (*λ_max_*) being in the range of 263–444 nm in the NMP solution and 300–354 nm in the polymer film (Table 3). In general, *λ_max_*s in polymer film were hipsochromically shifted (by about 11 and 27 nm) compared to the solution. Polyimides with azobenzene between imide rings exhibited a significant shift of the *λ_max_*s to the lower wavelength than polymers with azobenzene between amide linkages (Figure 6a). This shift can be connected with the formation of strong intermolecular H-bonds between the amide groups. The same blue shift was observed for azo poly(amide imide)s in our previous works [34,37,50]. The addition of azo dyes to the poly(amide imide) matrices caused the increasing absorbance at ca. 360 nm, but had no significant impact on the location of the *λ_max_* (Figure 6b; Appendix A). In the case of **PAI-1[Az(H)]**, a new absorption band in the range of 500–700 nm was observed (Appendix A).

### 2.5. Photoinduced Birefringence

The photoinduced birefringence in newly prepared materials was investigated using a 405 nm excitation beam of a moderate intensity of 100 mW/cm^2^. Among newly synthesized polyimides the beam was the most strongly absorbed by the layers of **PAI-2** and its guest-host analogue (Table 4). For these materials, the excitation wavelength was located close to the region of the absorption band maximum (Figure 6b). The lowest absorption coefficient at 405 nm (*α_405_*) was observed for **PAI-3**, for which the writing wavelength fell at the red shoulder of the polymer absorption band (Figure 6a). Considering the polyimides reported previously, **PAI-7** and **PAI-8** exhibited the highest *α_405_*_._ The measured changes in the photoinduced birefringence with an irradiation time of 405 nm are shown in Figure 7a. Figure 7b presents the birefringence decays recorded after turning off the excitation light. The maximum observed birefringence *(Δn_max_*) as well as a percentage of the relaxed birefringence *(Δn_relax_)*, due to the turning off of the excitation beam, together with the thickness of the polymer layers (*d*) used in the optical measurement, are summarized in Table 4.

The performed experiments revealed quite a different photoresponse of the investigated materials. Among the newly synthesized functionalized matrices, a large birefringence ~0.05 was induced in the polymers with the azo moieties between the amide groups (i.e., **PAI-2** and **PAI-1**), while a very low or zero birefringence was found for the matrices with the azo moieties between the imide rings (i.e., **PAI-5** and **PAI-3**) (Figure 7a). The former result may be attributed to both the efficient absorption of a 405 nm beam in the **PAI-2** and **PAI-1** layers and the highly inefficient disordering processes evidenced in their birefringence relaxation curves (Figure 7b). No birefringence generation in **PAI-3** indicated an insufficient free volume for azo chromophores to isomerize [34]. This hypothesis found a confirmation in the measured FTIR spectra of **PAI-3** (Figure 2), which revealed a large amount of intermolecular hydrogen bonds between the amide groups. Moreover, the lack of optical response of **PAI-3** is in agreement with a formation of interchain hydrogen bonds as well as a suppressed process of birefringence induction in **PAI-6**, i.e., in a structurally similar azo polymer to **PAI-3** [34]. A weak birefringence signal found for **PAI-5** further confirmed that the location of azo moieties exclusively between the imide groups was highly disadvantageous for a generation of photoinduced anisotropy. The non-zero signal observed in **PAI-5** in comparison to zero birefringence in **PAI-3** and **PAI-6** might be ascribed to differences in the polymer molar masses (low molar masses of **PAI-3** and **PAI-6** could favor the formation of a more extensive interchain H-bond).

It is worth noting that the stability of light-induced birefringence in both functionalized **PAI-1** and **PAI-2** polymers was particularly high and comparable to the birefringence stability reported for the **PAI-7** and **PAI-8** [35]. We recently demonstrated that the photoinduced properties of **PAI-7**, i.e., the magnitude of the generated birefringence and its stability after irradiation, allowed for an inscription of a complex azo chromophore pattern due to irradiation with two right-and-left circularly polarized coherent beams [51]. The irradiated **PAI-7** layers were successfully used as substrates of a liquid crystal cell that formed a switchable diffraction grating with unique diffraction properties. Thus, one can assume that the **PAI-1** and **PAI-2** materials showing similar photoresponsive behavior to **PAI-7**, exhibited a large application potential for the fabrication of liquid crystal-based optical devices.

Considering the optical response of the newly prepared guest-host azo systems, some interesting features may be noted. In particular, a birefringence signal that was detected for **PAI-3[Az(H)]** indicated that the addition of the **Az(H)** chromophores to the **PAI-3** matrix increased the distance between the polymer chains, which in turn allowed for trans-cis isomerization processes. However, as can be seen in Figure 7b, the molecular order induced in **PAI-3[Az(H)]** was unstable and almost fully relaxed within ca. 400 s after 405 nm exposure (Table 4). A very low birefringence and its fast decrease in the dark indicated that the non-bonded chromophores, rather than the covalently attached ones, made a major contribution to the observed optical response of the **PAI-3[Az(H)]** material.

The final birefringence of the guest-host **PAI-1[Az(H)]** and **PAI-2[Az(CH_3_)]** azo systems, was lower than or similar to the *Δn_max_* measured for their functionalized counterparts, respectively. However, the dynamics of birefringence growth in the **PAI-1[Az(H)]** and **PAI-2[Az(CH_3_)]** layers, during the first few hundreds of seconds of 405 nm exposure, were faster than in the covalent analogues. Interestingly, the birefringence stability of **PAI-2[Az(CH_3_)]** was as large as that observed in **PAI-2**, which may indicate an advantageous role of the methyl group substituent in stabilizing the induced molecular order. The birefringence signals did not reach saturation within the experimental time-period for either of the newly prepared polymers.

## 3. Materials and Methods

### 3.1. Materials

Trimetilic anhydride chloride, 2,6–diaminopyridine, pyridine, acetone, acetic anhydride, N–methyl–2–pyrrolidone (NMP), 1,2–dichlorobenzene, dimethyl sulfoxide (DMSO) and cyclohexanone were purchased from Sigma-Aldrich Chemical Co. Methanol, N,N–Dimethylformamide (DMF) and chloroform were purchased from Chempur (Poland). Tetrahydrofuran (THF) was purchased from Fluka (USA).

### 3.2. Measurements

The ^1^H NMR spectra were recorded on an Avance II 600 MHz Ultra Shield Plus (Bruker) Spectrometer in the DMSO-*d*_6_ as the solvent and with TMS as the internal standard. The infrared (IR) spectra of the polymers were recorded with a Nicolet 6700 FTIR apparatus (Thermo Scientific) with KBr pellets. The FTIR spectra of the guest-host azo systems were acquired on a BiO-Rad FTS 40 A Spectrometer in the transmission mode, using a polymer solution cast on the KBr pellets from the NMP solution. Prepared KBr pallets with non-covalent polyimides were dried in the vacuum at 50 °C by 24 h. The UV-Vis spectra were recorded as films on glass substrates and in NMP (solution 10^−5^ molL^−1^) of polymers using a V-570 UV-Vis-NIR Spectrophotometer (Jasco Inc.). The X-ray diffraction pattern of solid samples was recorded using Cu Kα radiation on a wide-angle HZG-4 diffractometer (Carl Zeiss Jena) working in the typical Bragg geometry. Thermogravimetric analysis (TGA) was performed on a Perkin Elmer Thermogravimetric Analyzer Pyris 1 TGA under a nitrogen atmosphere using heating/cooling cycles in the temperature range from 20–800 °C at a heating rate of 15 °C/min. Differential scanning calorimetry (DSC) was performed with a TA-DSC 2010 apparatus (TA Instruments) in a temperature range from 25–300 °C at a heating rate of 20 °C/min under a nitrogen flow of 50 mLmin^−1^. All samples for TGA and DSC analyses were dried before the measurement at 70 °C (non-covalent azo systems) or 100 °C (functionalized azo poly(amide imide)s) by 24 h. The molar mass and dispersity (*M_w_/M_n_*) of the polymers were determined using gel permeation chromatography (GPC) measurements conducted at room temperature with DMF as an eluent at a flow rate of 1 mLmin^−1^. A Knauer apparatus with MIXED- DPL gel columns (Knauer) and polystyrene standards were used.

Photoinduced birefringence (*Δn**)* measurements were carried out using a 405 nm excitation beam and a 690 nm probe beam from diode lasers. The detailed scheme of an experimental set-up was presented elsewhere [33]. The technique was based on a detection of the probe beam transmission through a polymer sample placed between two crossed polarizers, while irradiating the sample with the excitation beam. In the measurement, the polarization direction of the excitation beam was oriented at an angle of 45° with respect to a polarization of the probing beam. For such an experimental configuration, the photoinduced birefringence was calculated from the formula: *Δn =*
*λ/(**πd)arcsinT*^1/2^, where *λ* is the probing beam wavelength, *d* is the film thickness and *T* is the transmittance of the crossed-polarizer set-up. The measurements of the thickness of the poly(amide imide) layers cast on glass substrates were performed by a Dektak XT stylus profiler with a diamond tipped stylus.

### 3.3. Synthesis of Azo Chromophores 

Azo chromophores 4-[(4-methyl phenyl)diazenyl]phenol (**Az(CH_3_)**) and 4-phenylazophenol (**Az(H)**) were synthesized and were comprehensively characteristic as they were in articles [25,31].

### 3.4. Synthesis of Diamines 

The detailed synthesis and characterization of the chromophores 2,4-diamino-4’-methylazobenzene (**AKCH_3_**) and 2,4-diamino-4’-azobenzene (**AK(H)**) have been previously reported [21,52].

### 3.5. Synthesis of Dianhydrides

The detailed synthesis and characterization of the azo dianhydrides 4,4’-(2,4-diamido-azobenzene)bisphthalic anhydride (**DB(H)**), 1,3-[pyridine-di(benzene-5-amido-1,2-dicarboxylic)dianhydride (**DB(Py)**) and 4,4’-(2,4-diamido-4-methylazobenzene)bisphthalic anhydride (**DB(CH_3_)**) have been previously reported [1,36,53].

### 3.6. Synthesis of Chromophore-Functionalized Polymers

The synthesis of the azo poly(amide imide)s with pyridine rings in the main-chain was carried out according to the published procedure [54]. In a 25 mL round bottom flask, equimolar amounts of dianhydride (1 mmol) and the appropriate diamine (1 mmol) were placed. The whole was dissolved in a solvent mixture of 2.4 mL of NMP and 0.6 mL of 1,2-dichlorobenzene. O-dichlorobenzene was added as a co-solvent to remove the water formed during the imidization process. The reaction mixture was heated for 3.5 h on a magnetic stirrer at 175 °C in an argon atmosphere. The product was precipitated in methanol, purified by methanol extraction on a Soxhlet apparatus for 2 days and dried at 100 °C.

**PAI-1** was synthesized from azo dianhydride **DB(H)** and diamine 2,6-diaminopyridine. Yield: 95%. ^1^H NMR (DMSO-*d*_6_, 600 MHz, δ, ppm): 7.58 (s, ArH, 3H), 7.79 (s, ArH, 1H), 7.93 (s, ArH, 5H), 8.20 (s, ArH, 3H), 8.36 (s, ArH, 1H), 8.53 (s, ArH, 2H), 8.57–8.66 (d, ArH, 1H), 8.86 (s, ArH, 1H), 11.07 (s, NH, 1H), 11.39 (s, NH, 1H). FTIR (KBr, cm^−1^): 3353 (N–H in amide group); 1784, 1729 (C=O in imide group); 1679 (C=O in amide group); 1599 (–N=N–); 1356 (–C–N– stretching); 1001 (pyridine); 720 (–C–N– deformation). Anal. Calcd. (%) for [C_35_H_19_N_7_O_6_]_n_ (633.57 g/mol): C, 66.35; H, 3.02; N, 15.48: Found: C, 66.56; H, 3.37; N, 15.94.

**PAI-2** was synthesized from azo dianhydride **DB(CH_3_)** and diamine 2,6–diaminopyridine. Yield: 85%. ^1^H NMR (DMSO-*d*_6_, 600 MHz, δ, ppm): 2.38 (s, CH_3_, 3H), 7.38 (s, ArH, 3H), 7.7–7.87 (d, ArH, 4H), 7.89 (s, ArH, 1H), 8.19 (s, ArH, 3H), 8.37 (s, ArH, 1H), 8.53 (s, ArH, 2H), 8.57–8.66 (d, ArH, 1H), 8.87 (s, ArH, 1H), 11.04 (s, NH, 1H), 11.41 (s, NH, 1H). FTIR (KBr, cm^−1^): 3341 (N–H in amide group); 3053 (CH_3_); 1783, 1728 (C=O in imide group); 1679 (C=O in amide group); 1597 (–N=N–); 1356 (–C–N– stretching); 1000 (pyridine); 719 (–C–N– deformation). Anal. Calcd. (%) for [C_36_H_21_N_7_O_6_]_n_ (647.59 g/mol): C, 66.77; H, 3.27; N, 15.14:. Found: C, 66.04; H, 3.99; N, 14.50.

**PAI-3** was synthesized from azo dianhydride **DB(Py)** and diamine **AK(H)**. Yield: 37%. ^1^H NMR (DMSO-*d*_6_, 600 MHz, δ, ppm): 6.89 (d, ArH, 2H), 7.24 (d, ArH, 2H), 7.94 (s, ArH, 3H), 8.05–8.10 (d, ArH, 2H), 8.10-8.15 (d, ArH, 2H), 8.44 (s, ArH, 1H), 8.46 (s, ArH, 2H), 8.49 (s, ArH, 1H), 8.51 (s, ArH, 1H), 8.56 (s, ArH, 1H), 11.11 (d, NH, 2H). FTIR (KBr, cm^−1^): 3151 (N-H); 1781, 1715 (C=O in imide group); 1658 (C=O in amide group); 1603 (–N=N–); 1380 (–C–N– stretching); 1000 (pyridine); 723 (–C–N– deformation). Anal. Calcd. (%) for [C_35_H_19_N_7_O_6_]_n_ (633.57 g/mol): C, 66.35; H, 3.02; N, 15.47:. Found: C, 65.94; H, 3.51; N, 14.92. 

### 3.7. Preparation of Non-Covalent Azo Polyimides

The non-covalent azo systems (**PAI-1[Az(H)]**, **PAI-2[Az(CH_3_)]** and **PAI-3[Az(H)]**) were prepared by the dissolution of the chromophores guest and azo poly(amide imide)s hosts in NMP solution. Each system contained equimolar amounts of the matrix and chromophores. The non-covalent azo polymers were prepared in the form of films on glass substrates.

### 3.8. Polymer Film Preparation

The homogenous solutions of 0.1 g of functionalized polyimides (**PAI-1**, **PAI-2**, **PAI-3**) or the non-covalent azo systems (**PAI-1[Az(H)]**, **PAI-2[Az(CH_3_)]** and **PAI-3[Az(H)]**) were dissolved in 1.5 mL NMP and filtered through 0.2 µm membranes. Then, they were applied on to microscopic glass substrate and heated on a magnetic stirrer at 50 °C. Subsequently, the layers were heated in a vacuum oven at 70 °C for 3 days (non-covalent azo systems) or 100 °C for 24 h (functionalized polyimides) to completely evaporate the solvent.

## 4. Conclusions

The structure-property relations of functionalized “T-type” poly(amide imide)s and guest-host analogues were compared using DSC, TGA, UV-Vis and photoinduced birefringence measurements. Our results showed the increase of *T_g_* for functionalized polyimides with pyridine rings in comparison to analogues with phenyl rings. It was noticed that polymers containing covalently bonded chromophores exhibit better thermal properties, than their analogues with the doped azo dyes. A strong dependence of the absorption band and the location of the chromophores in the backbone was observed. Functionalized azo polyimides with chromophores between imide rings showed a large bathochromic shift of the *λ_max_* in comparison to the polymers containing azo moieties between amide linkages. The red-shift can be connected to the formation of the strong intermolecular H-bonds, which was evidenced by the FTIR and ^1^H NMR spectra. The spectroscopic analysis showed that the presence of amide and imide groups in the main-chain of the polymer backbone hinders the formation of the H-bonds between nitrogen in pyridine rings and the hydroxyl group in the azo chromophores. Our present and former results of photoinduced birefringence measurements revealed important relationships between the chemical structure and photoresponsive behavior of functionalized “T-type” azo poly(amide imide)s. Firstly, a location of azo chromophores between the imide rings instead of the amide groups is highly disadvantageous for the efficient absorption of blue light as well as for the effective trans-cis photoisomerization; a larger molar mass of the polymers with azo moieties between the imide rings or introducing azo chromophore dopants may have an advantageous effect on the detection of the non-zero optical response. Secondly, for the efficient process of photoinduced birefringence generation, the presence of azo chromophores between the amide groups is necessary. In that case, the induced birefringence exhibits a very large stability after irradiation, making these particular azo polymers very attractive for applications in the area of the photoalignment of liquid crystals. It should be noticed that our comparative studies (current and previous) allowed the formulation of some structure-properties relationships that can be applied to the entire class of “T-type” azo poly(amide imide)s.

## Figures and Tables

**Figure 1 materials-13-01912-f001:**
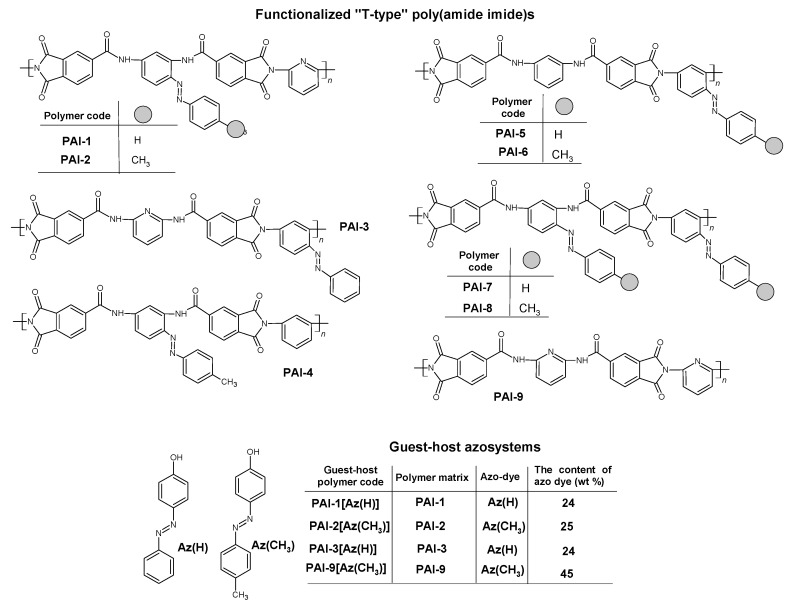
Chemical structures of the functionalized poly(amide imide)s, azo chromophores and the guest-host systems with the content of azo dye.

**Figure 2 materials-13-01912-f002:**
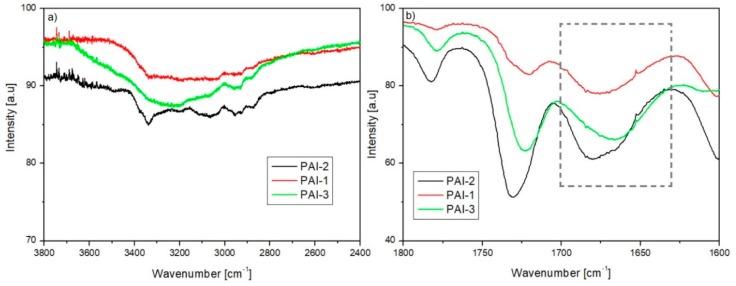
FTIR spectra of the azo polyimide matrices **PAI-1**, **PAI-2**, **PAI-3** in the range of (**a**) 2400–3800 cm^−1^ and (**b**) 1600–1800 cm^−1^.

**Figure 3 materials-13-01912-f003:**
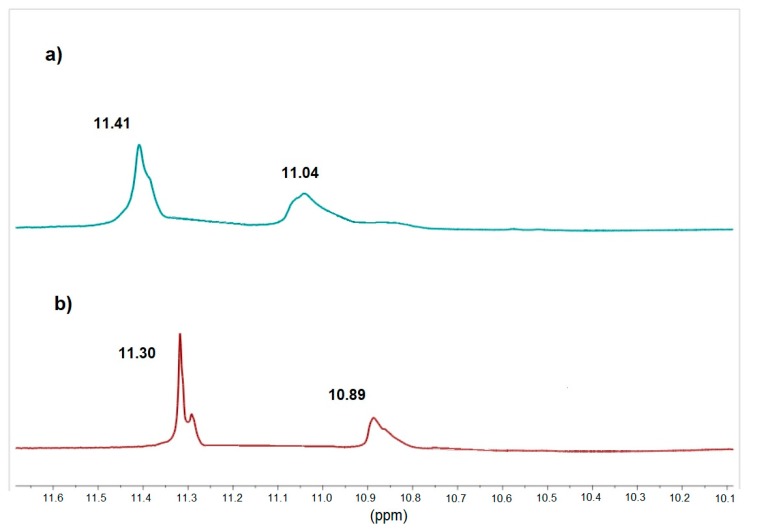
^1^H NMR (DMSO-*d*_6_) spectra of **PAI-2** (**a**) at room temperature and (**b**) at 80 °C.

**Figure 4 materials-13-01912-f004:**
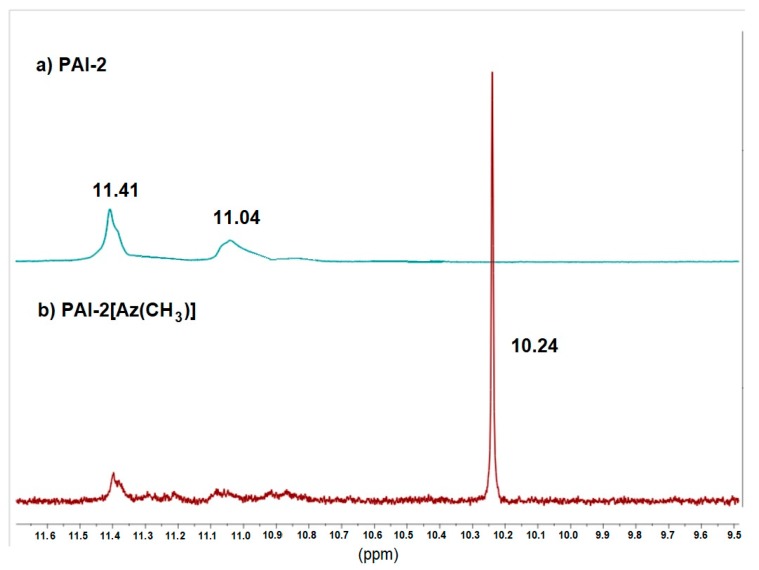
The ^1^H NMR spectra for (**a**) the polyimide matrix **PAI-2** and (**b**) its guest-host azo system analogue **PAI-2[Az(CH_3_)]**.

**Figure 5 materials-13-01912-f005:**
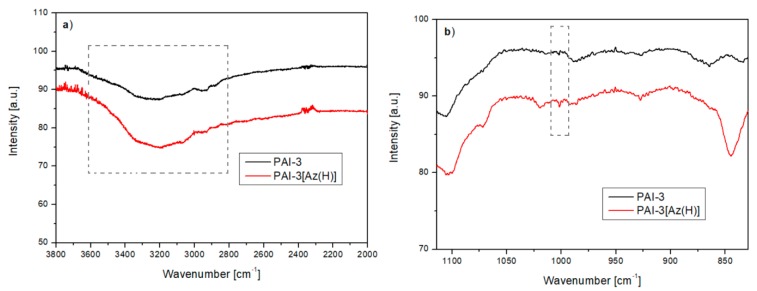
FTIR spectra for the polyimide matrix **PAI-3** and their guest-host azo system analogue **PAI-3[Az(H)]** in the range of (**a**) 2000–3800 cm^−1^ and (**b**) 800–1250 cm^−1^.

**Figure 6 materials-13-01912-f006:**
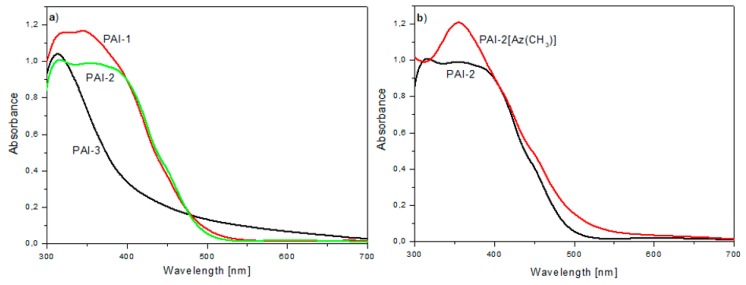
UV-Vis spectra of (**a**) the azo polyimide matrices and (**b**) the **PAI-2[Az(CH_3_)]** compared with **PAI-2** in the polymer film.

**Figure 7 materials-13-01912-f007:**
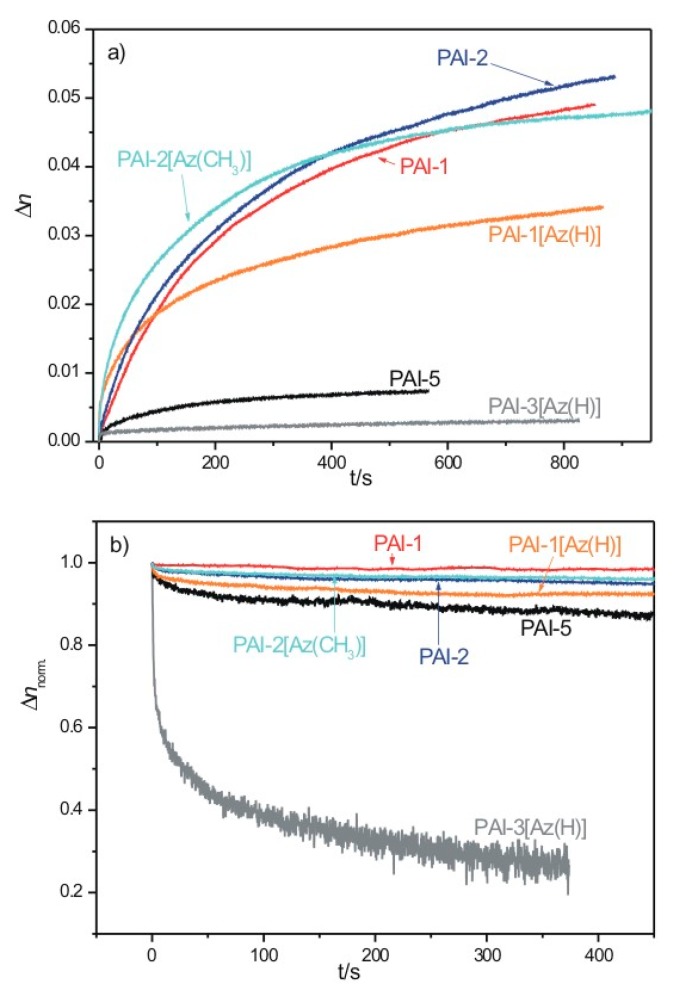
(**a**) Photoinduced birefringence growths upon 405 nm 100 mW/cm^2^ excitation beam (**b**) the corresponding normalized birefringence decreases after turning off the excitation beam in the studied azo polyimides.

**Table 1 materials-13-01912-t001:** Molar masses and dispersity of the functionalized poly(amide imide)s.

Polymer Code	*M_n_* × 10^3^ (g/mol)	*M_w_* × 10^3^ (g/mol)	*M_w_/M_n_*
PAI-1	3.2	6.2	1.9
PAI-2	3.2	5.2	1.7
PAI-3	1.1	1.7	1.5
PAI-5 [37]	2.1	3.85	1.8
PAI-6 [34,35]	1.1	1.9	1.7
PAI-7 [35]	4.3	6.8	1.6
PAI-8 [34,35]	10.3	14.9	1.5

**Table 2 materials-13-01912-t002:** Glass transition temperatures (*T_g_*) and the thermal stability of functionalized poly(amide imide)s, azo chromophores and the guest-host azo systems.

Polymer Code	DSC	TGA (N_2_)			
	*T_g_* (°C)	^a^*T_5_* (°C)	^b^*T_10_* (°C)	^c^*T_max_* (°C)	^d^ Residual Weight (%)
PAI-1	244	419	455	432; 612	53
PAI-2	281	426	489	427; 596	54
PAI-3	300	357	416	390; 533	49
PAI-5	282	331	408	356; 463; 552	53
PAI-6	247 [35]	290 [34,35]	407	434; 598	50
PAI-7 [35]	254	371	412	429; 606	49
PAI-8 [33,34,35]	nd	258	389	-	33
PAI-9 [36]	265	356	412	454; 565	69
PAI-1[Az(H)]	198	202	221	214; 437; 593	38
PAI-2[Az(CH_3_)]	210	211	235	226; 405; 581	42
PAI-3[Az(H)]	133	210	235	225; 389; 597	45
PAI-9[Az(CH_3_)] [36]	105	174	229	172; 362	40
Az(H) [47]	−73	188	202	244	-
Az(CH_3_) [47]	4	114	124	127; 237; 289	-

nd—not detected up to 300 °C; ^a^ Temperature of 5% weight loss; ^b^ Temperature of 10% weight loss; ^c^ Temperature of the maximum decomposition rate from the differential thermogravimetric curves (DTG); ^d^ Residual weight at 800 °C.

**Table 3 materials-13-01912-t003:** Maximum absorption wavelength (*λ_max_*) of the functionalized polyimides, the azo systems in film and the N-methyl-2-pyrrolidone (NMP) solution.

Polymer Code	*λ_max_* (nm) in Film	*λ_max_* (nm) in NMP
PAI-1	317; 345	311; 361
PAI-2	314; 354	308; 381
PAI-3	310	314
PAI-5 [37]	below 300	263; 311; 444 *
PAI-6 [34,35]	309	264
PAI-7 [33]	342	292 *; 338; 395 *; 440 *
PAI-8 [33]	355	263; 350; 447 *
PAI-9 [36]	323	301
PAI-1[Az(H)]	349	
PAI-2[Az(CH_3_)]	355	
PAI-3[Az(H)]	343	
PAI-9[Az(CH_3_)] [36]	353	

* The position of the absorption band calculated using the second derivatives method (i.e., the minimum of the second derivative of the absorption curve corresponds to the absorption maximum).

**Table 4 materials-13-01912-t004:** The absorption coefficient at 405 nm, the thickness, maximum induced birefringence and the percentage of the relaxed birefringence measured for the polymer layers cast onto glass substrates.

Polymer Code	*α_405_* (cm^−1^)	*d* (µm)	*Δn_max_*	*Δn_relax_* (%)
PAI-1	5.8 × 10^4^	0.40	0.049	2
PAI-2	8.1 × 10^4^	0.25	0.052	5
PAI-3	2.7 × 10^4^	0.37	non-detectable	-
PAI-5	3.0 × 10^4^	0.50	0.007	12
PAI-6 [34,35]	3.0 × 10^4^	0.38	non-detectable	-
PAI-7 [35]	1.0 × 10^5^	0.19	0.055 (after 600 s)	1.5 (after 500 s)
PAI-8 [34,35]	1.0 × 10^5^	0.13	0.060 (after 1000 s)	5 (after 500 s)
PAI-1[Az(H)]	7.2 × 10^4^	0.25	0.034	8
PAI-2[Az(CH_3_)]	8.5 × 10^4^	0.25	0.047	4
PAI-3[Az(H)]	3.5 × 10^4^	0.61	0.003	73
PAI-9[Az(CH_3_)] [36]	3.1 × 10^4^	0.35	0.012 (after 600 s)	25 (after 500 s)

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
