# Peer review of "Azobenzene Functionalized “T-Type” Poly(Amide Imide)s vs. Guest-Host Systems—A Comparative Study of Structure-Property Relations"

_materials, 2020, doi:10.3390/ma13081912_

Round 1
Reviewer 1 Report
The authors report synthesis and characterization of four functional azobenzene polymers for photonics applications. The newly synthesized polymers are compared against each other and other similar polymers previously synthesized and reported by the authors. While somewhat incremental, the manuscript is clear and well written, and I have only minor comments and suggestions.
While “T-type” is rather well understood within the context of the paper, a sentence or two would not hurt to introduce the principal ideas compared to main chain and side chain polymers.
There are a few acronyms; POA, SRG, DR1 used only two times which makes introducing them rather unnecessary. “PI” is left unexplained.
Line 260-261: While PAI-2 indeed has the highest absorption coefficient of the newly synthesized polymers, the fact that PAI-7 and PAI-8 have yet higher absorbance, as shown in table 4, should not be neglected.
Table 4: The used symbols for different physical properties should be included in the caption. The “relaxed birefringence” should be opened in the text. Now it is unclear whether it is the remaining portion or the portion that vanished with time and the reader needs to deduce this information from figure 7.
Lines 289-291: The fact that no birefringence is seen for the polymers with the lowest molecular weights (1100 g/mol) and the lowest measurable is found for the polymer with the second-lowest (2100 g/mol) deserves a bit more consideration: Is it certain that the low molecular weight suggesting oligomeric structure versus a true polymer chain (as properly mentioned by the authors earlier in the text) cannot be the main thing to explain the very low levels of photoinduced birefringence?
A few typos or inconsistencies in the text:
Line 86 and 178-183: font size
Line 223: “Incorporation pyridine ring…”
Line 229: Should “highest increase” read “highest decrease”?
Figure 6: “Wavelnegth”
Supplementary references and supplementary content:
The related terms TGA, DTA and DTG are used rather mixed. Do the authors call figure S2 DTG or DTA curve?
Fig. S2. What is the y-axis reading in units of inverse temperature? For differential thermal analysis I would expect y-axis to be temperature difference and for differential thermogravimetry weight loss rate.
GPC is said to have been performed at 800 °C temperature. Does not seem plausible.
More detail should be given on the fabrication of the sample films. How was even film thickness assured and, more importantly, how were the film thickness measurements performed? Accurate film thickness is very important for accurate comparison of the photoinduced birefringence.
Reviewer 2 Report
The authors in this manuscript reported a comparative study on the structure-property relations of Azobenzene functionalized “T-type” poly(amide imide)s and “guest host” systems through various techniques like 1H NMR spectroscopy, FTIR spectroscopy, X-ray diffraction, DSC, TGA, UV-Vis spectroscopy and photoinduced birefringence measurements. And revealed the relationships between functionalized poly(amide imide)s, azo chromophores, and guest-host azo systems and their thermal and optical properties. Technically, the authors did very good characterization of these materials, but at current stage this manuscript is still premature due to following issues.
1. The authors provided Materials and Methods section in their supplementary materials, but I would strongly suggest the authors move this part to the main text in line with the “Instructions for Authors” of this journal.
2.The authors mentioned 11 times of their previous works throughout this manuscript, which gave readers an impression that they are not reading something novel. As a new article, it would be beneficial to readers if the authors can avoid the redundant words and use more concise texts to describe their former results.
3. The authors need to define some acronyms. Like “PI” In line 55, “DR1” in line 56.
4. In Table 3, the authors gave the λmax of PAI-5 and PAI-6, but didn’t provide the corresponding UV-vis spectra.
5. In Figure 5, the authors should also mention the dip for PAI-3[Az(H)] at 850 cm-1 in discussion.
6.Other minors: carefully check the font size and style starting from line 178, which is different with the other parts; no index for PAI-2 in figure 4 1H NMR spectra.
Therefore, this manuscript can be considered for publication until the above comments are addressed satisfactorily.
Reviewer 3 Report
This work offers a comparative study of structure-property relations between azobene functionalized “t-type” poly(amide imide) and guest-host systems. This is a very interesting scientific contribution, particularly from the point of view of the development of optoelectronic devices. The paper is very well structured, exhibiting a valuable set of conclusions properly supported by the work performed throughout the text. I therefore recommend this work to be considered for acceptance in MDPI Materials, but just after performing a quick review in the form of minor revisions. I include some brief comments hereby:
1.- In the Introduction section, some applications of new materials in photonic processes are mentioned (lines 25-28). However, no reference is directly found in this paragraph to optical fiber sensing and polymer sensitive coatings, which is a topic gathering lot of interest lately. I recommend to include a mention to optical sensing and optical fiber sensing related to polymer materials, as well as the following references: - Sensors and Actuators B: Chemical 244, 1145-1151 (2017)
- Sensors and Actuators B: Chemical 254, 1087-1093 (2018)
2.- Line 86: The text format is altered. Please, correct it.
3.- Figure 4: The peak value corresponding to the polymide matrix PAI-2 (blue trace) is missing. Please, insert it. Additionally, could you increase the resolution of the figure? The reason is that the horizontal axis cannot be read properly.
4.- Line 158-159: Delete new line spacing.
5.- Lines 177-183: The text format is altered. Please, correct it.
6.- Fig 6: Could authors apply the same scale in the vertical axis of both figures so that the figures can be easier compared?
Reviewer 4 Report
Comment 1
“…..π-π sticks” a recent work on π-π interaction towards increased polymer – filler interaction using new grafting polymer is suggested to cite: Polymer 148 (2018) 247-258 https://doi.org/10.1016/j.polymer.2018.06.025
Comment 2
Line 117 “The comparison between the obtained spectra shows that the N-H absorption region at ca. 3300 cm-1 is higher in intensity and broaden for PAI-3 (Fig. 2a).”
Are FTIR spectras normalized? If yes mention so in the manuscript, if not please justify the first half of this sentence
Comment 3
Line 195 “….. required a proper drying procedure…” please mention the conditions used in the work
Round 2
Reviewer 2 Report
I have gone through the revised manuscript. The authors have answered all questions raised by this reviewer and made corresponding changes in the manuscript. The explanations sound reasonable and acceptable based on the further evidences. This revised paper looks in good shape and is well organized after text amendments. This reviewer is therefore convinced that the work can be published in this journal.